# Adapter Pruning using Tropical Characterization

**Rishabh Bhardwaj**[⬚]   **Tushar Vaidya**[⬚]   **Soujanya Poria**[⬚]

[⬚] Singapore University of Technology and Design, Singapore

[⬚] Nanyang Technological University, Singapore

rishabh_bhardwaj@mymail.sutd.edu.sg, tushar.vaidya@ntu.edu.sg

sporia@sutd.edu.sg

## Abstract

Adapters are widely popular parameter-efficient transfer learning approaches in natural language processing that insert trainable modules in between layers of a pre-trained language model. Apart from several heuristics, however, there has been a lack of studies analyzing the optimal number of adapter parameters needed for downstream applications. In this paper, we propose an adapter pruning approach by studying the tropical characteristics of trainable modules. We cast it as an optimization problem that aims to prune parameters from the adapter layers without changing the orientation of underlying tropical hypersurfaces. Our experiments on five NLP datasets show that tropical geometry tends to identify more relevant parameters to prune when compared with the magnitude-based baseline, while a combined approach works best across the tasks.

## 1 Introduction

With the increase in network sizes, we are observing an ever-increasing space and computational demand for models needed to solve a given task. To tackle this, model compression (Cheng et al., 2017) techniques are becoming continuously popular which retain the most important learning from the full model while reducing the size of the network either by pruning or distillation.

Transfer learning approaches, such as adapters (Houlsby et al., 2019), are a parameter-efficient alternative to full model fine-tuning which obviates the need to maintain a task-specific copy of the base language model (LM). Adapters insert simple modules in between layers of an LM to adapt the pre-trained representation for a given downstream NLP task. However, there is a lack of research in pruning adapter modules to further enhance their parameter efficiency. We hypothesize that adapter weights can be pruned significantly by not compromising the performance observed with unpruned states, this motivates the proposed approach.

In this work, we propose a novel approach to pruning adapter layers without any iterative fine-tuning of the model parameters on downstream tasks. Using tropical algebra, we study the (duals of) hypersurfaces generated by adapter modules in the high-dimensional space. As a pruning objective, we aim to minimize the magnitude of adapter weights while constraining the change in hypersurface geometry to be small.

Related works include adapters pruning using lottery ticket hypothesis (Wu et al., 2022; Frankle and Carbin, 2018) that performs iterative pruning—a few gradient steps, prune, and reset the parameters to initial weights. Rücklé et al. (2020) drops adapter from lower transformer layers. While these works are interesting, we provide a more concrete angle to prune adapter layers—prune by preserving the hypersurface geometry. We extend an insightful analysis of tropical geometry of neural networks (Zhang et al., 2018; Alfarra et al., 2022) to adapters.

## 2 Background

**Adapter Operations.**   We use the adapter setup proposed by Pfeiffer et al. (2020) that inserts small modules after FFN add and layer norm sub-layer.

$$\boldsymbol{h} \leftarrow \boldsymbol{h} + f(\boldsymbol{h}\boldsymbol{W}_d)\boldsymbol{W}_u \qquad (1)$$

It consists of down-projection $\boldsymbol{W}_d \in \mathbb{R}^{d \times r}$, up-projection $\boldsymbol{W}_u \in \mathbb{R}^{r \times d}$, a ReLU activation function $f(\cdot)$, where typically $r < d$.

**Tropical Arithmetic.**   Tropical algebra is a variant of classical algebra where basic arithmetic operations are redefined. The tropical sum $\oplus$ of two numbers represents their maximum and the tropical product $\odot$ represents a classical addition[1]. Thus,

$$x \oplus y = \max\{x, y\}$$
$$x \odot y = x + y$$

---

[1] The tropical addition can be defined as $a \oplus b = \min\{a, b\}$ or $\max\{a, b\}$, we focus on the latter as we analyze a ReLU-based adapter network.

For e.g., $2 \oplus 5 = 5$ and $2 \odot 5 = 7$. Axioms and order of arithmetic operations in tropical algebra follow the classical, thus addition is commutative and multiplication is distributive over addition. We relegate detailed discussions about tropical algebra, polynomials, and hypersurfaces to the Appendix A.

**Notations used:** Henceforth, we denote $\boldsymbol{W}_d$, $\boldsymbol{W}_u$, $\boldsymbol{h}$ by $\mathbf{A}$, $\mathbf{B}$, and $\mathbf{x}$, respectively; $\mathbf{B}^+ := \max\{\mathbf{B}, \mathbf{0}\}$; $\mathbf{B}^- := \max\{-\mathbf{B}, \mathbf{0}\}$; $b_i$ denotes $i_{th}$ row of $\mathbf{B}$; $\boldsymbol{b}^{i+} := \max\{\boldsymbol{b}^i, \mathbf{0}\}$, $\boldsymbol{b}^{i-} := \max\{-\boldsymbol{b}^i, \mathbf{0}\}$; $\mathrm{Diag}[\boldsymbol{u}]$ arranges $\boldsymbol{u}$ in a diagonal matrix; $||\mathbf{G}||_{1,1} := \Sigma_{k=1}^d ||\mathbf{G}(i,:)||_1$; $||\cdot||_F$ denotes Frobenius Norm.

## 3 Tropical Adapter Pruning

Given a frozen language model adapted to a specific task using adapter layers, we divide our approach into two steps: 1) Finding adapter weights $P_T$ that are crucial to preserving the tropical adapter hypersurface by solving a simple optimization problem; 2) Pruning of adapter weights with least magnitudes that do not lie in $P_T$. Next, we describe step-1 which is core to the pruning method:

A bottleneck adapter block can be expressed by $f(\mathbf{x}) = \mathbf{B}\max\{\mathbf{A}\mathbf{x}, \mathbf{0}\}$. Since $f(\mathbf{x})$ in itself is not a tropical polynomial and thus does not form a tropical surface, we rewrite it in terms of the difference between two tropical polynomials $f(\mathbf{x}) = H(\mathbf{x}) - Q(\mathbf{x})$, following the analysis of tropical rational function by Alfarra et al. (2022). Thus we focus on a relatively lenient problem i.e. identifying weights that preserve tropical hypersurfaces defined by $H(\mathbf{x})$ and $Q(\mathbf{x})$. Let $\mathcal{H}(\mathbf{x})$ and $\mathcal{Q}(\mathbf{x})$ be the respective hypersurfaces, one can choose a sparse set of $\hat{\mathbf{A}}, \hat{\mathbf{B}}$ that belongs to the set of matrices obtained by solving the following optimization problem

$$\min_{\hat{\mathbf{A}}, \hat{\mathbf{B}}} \quad d(\mathcal{H}(\mathbf{x}), \hat{\mathcal{H}}(\mathbf{x})) + d(\mathcal{Q}(\mathbf{x}), \hat{\mathcal{Q}}(\mathbf{x}))$$

Where $d(\cdot)$ defines the distance between two geometric objects; $\hat{\mathcal{H}}$ and $\hat{\mathcal{Q}}$ are hypersurfaces obtained by substituting $\mathbf{A}$ and $\mathbf{B}$ with $\hat{\mathbf{A}}$ and $\hat{\mathbf{B}}$ in $f(\mathbf{x})$. In place of preserving the orientation of $\mathcal{H}(\mathbf{x})$ and $\mathcal{Q}(\mathbf{x})$, we aim to preserve the orientation of their respective dual objects denoted by $\delta(H(\mathbf{x}))$ and $\delta(Q(\mathbf{x}))$. Thus,

$$\min_{\hat{\mathbf{A}}, \hat{\mathbf{B}}} d\Big(\delta(\mathcal{H}(\mathbf{x})), \delta(\hat{\mathcal{H}}(\mathbf{x}))\Big) + d\Big(\delta(\mathcal{Q}(\mathbf{x})), \delta(\hat{\mathcal{Q}}(\mathbf{x}))\Big)$$

Without the loss of generality, we assume down-projection is bias-free[2], $\delta(\cdot)$ can be expressed in terms of generator matrices $\mathbf{G}$ of zonotopes obtained from $\mathbf{A}$ and $\mathbf{B}$. To find sparse $\hat{\mathbf{A}}, \hat{\mathbf{B}}$, we introduce sparse regularization terms in the optimization function. Thus, finding adapter weights that preserve the hypersurface geometry can be cast as the following optimization problem:

$$\min_{\hat{\mathbf{A}}, \hat{\mathbf{B}}} \frac{1}{2}\Big|\Big|\hat{\mathbf{G}}_1 - \mathbf{G}_1\Big|\Big|_F^2 + \frac{1}{2}\Big|\Big|\hat{\mathbf{G}}_2 - \mathbf{G}_2\Big|\Big|_F^2$$
$$+ \lambda_1\Big|\Big|\hat{\mathbf{G}}_1\Big|\Big|_{1,1} + \lambda_2\Big|\Big|\hat{\mathbf{G}}_2\Big|\Big|_{1,1} \quad (2)$$

where $\mathbf{G}_1 = \mathrm{Diag}[\boldsymbol{b}^{i+}]\mathbf{A}$; $\mathbf{G}_2 = \mathrm{Diag}[\boldsymbol{b}^{i-}]\mathbf{A}$,
$\hat{\mathbf{G}}_1 = \mathrm{Diag}[\hat{\boldsymbol{b}}^{i+}]\hat{\mathbf{A}}$; $\hat{\mathbf{G}}_2 = \mathrm{Diag}[\hat{\boldsymbol{b}}^{i-}]\hat{\mathbf{A}}$,

We provide a derivation of the above function in Appendix B. It is important to note that in the pruning phase, we do not iteratively fine-tune adapter or LM parameters on the downstream task.

---

**Algorithm 1:** Tropical Adapter Pruning

**Initialize:** $T, \eta$ $\lambda_1, \lambda_2$;
**Return:** $\hat{\mathbf{A}}, \hat{\mathbf{B}}$;
$\hat{\mathbf{B}}^+ \leftarrow \mathbf{B}^+, \hat{\mathbf{B}}^- \leftarrow \mathbf{B}^-$;
**for** *t in 1,..., T* **do**
  **for** *i in 1,..., r* **do**
    **if** *t is even* **then**
      $\hat{\mathbf{G}}_1^i = \mathrm{Diag}[\hat{\boldsymbol{b}}^{i+}]\hat{\mathbf{A}}$;
      $loss_1 = ||\hat{\mathbf{G}}_1^i - \mathbf{G}_1^i||_F^2$;
      $loss_2 = ||\hat{\mathbf{G}}_1||_{1,1}$;
      $\ell = 0.5 * loss_1 + \lambda_1 * loss_2$;
    **else**
      $\hat{\mathbf{G}}_2^i = \mathrm{Diag}[\hat{\boldsymbol{b}}^{i-}]\hat{\mathbf{A}}$;
      $loss_1 = ||\hat{\mathbf{G}}_2^i - \mathbf{G}_2^i||_F^2$;
      $loss_2 = ||\hat{\mathbf{G}}_2||_{1,1}$;
      $\ell = 0.5 * loss_1 + \lambda_2 * loss_2$;
    **end**
    < check convergence of combined loss >
    $\hat{\mathbf{A}} \leftarrow \hat{\mathbf{A}} - \eta * \frac{\partial}{\partial(\hat{\mathbf{A}})}\ell$;
    $\hat{\mathbf{B}} \leftarrow \hat{\mathbf{B}} - \eta * \frac{\partial}{\partial(\hat{\mathbf{B}})}\ell$.
  **end**
**end**

---

[2]We merge bias term $\mathbf{b}$ with the down-projection matrix, thus $\mathbf{x} \leftarrow [\mathbf{x}, 1]$ and $\mathbf{A} \leftarrow [\mathbf{A}; \mathbf{b}]$.

| | 1.4% | 2.9% | 5.7% | 8.6% | 11.5% | 14.5% | 17.4% | 20.3% | 23.3% | FM |
|---|---|---|---|---|---|---|---|---|---|---|
| | | | | | Task-1 (MELD) | | | | | |
| Standard | 33.56 | 35.00 | 41.02* | 39.65 | 43.72 | 49.38 | 50.18 | 55.05 | 57.06 | |
| Tropical | 34.06* | 37.37* | 36.79 | 49.59* | 52.46* | 52.28* | 56.34* | 58.04* | 58.09* | 60.71 |
| Combined | 33.56 | 37.37 | 41.02 | 49.59 | 52.46 | 52.28 | 56.34 | 58.04 | 58.09 | |

| | 1.4% | 2.9% | 4.3% | 11.5% | 13.0% | 14.5% | 26.0% | 32.8% | 39.4% | FM |
|---|---|---|---|---|---|---|---|---|---|---|
| | | | | | Task-2 (SNLI) | | | | | |
| Standard | 40.31 | 34.41 | 34.74 | 77.91* | 79.96* | 82.06* | 85.77 | 86.01 | 85.77 | |
| Tropical | 41.08* | 46.33* | 46.94* | 75.82 | 77.93 | 78.17 | 85.96* | 86.17* | 85.96* | 86.45 |
| Combined | 41.08 | 46.33 | 46.94 | 77.91 | 79.96 | 82.06 | 85.96 | 86.01 | 85.96 | |

| | 1.4% | 2.7% | 4.2% | 5.8% | 7.3% | 8.9% | 11.1% | 12.1% | 15.0% | FM |
|---|---|---|---|---|---|---|---|---|---|---|
| | | | | | Task-3 (RT) | | | | | |
| Standard | 77.49 | 74.67 | 69.23 | 82.65 | 83.86 | 86.49 | 83.02 | 85.18 | 87.80* | |
| Tropical | 79.17* | 82.55* | 83.68* | 84.05* | 86.12* | 87.05* | 87.24* | 88.37* | 87.71 | 87.99 |
| Combined | 79.17 | 82.55 | 83.68 | 84.05 | 86.12 | 86.49 | 87.24 | 88.37 | 87.71 | |

| | 1.4% | 2.9% | 5.7% | 8.5% | 11.5% | 14.4% | 17.4% | 20.3% | 26.3% | FM |
|---|---|---|---|---|---|---|---|---|---|---|
| | | | | | Task-4 (IMDB) | | | | | |
| Standard | 74.55 | 71.70 | 69.82 | 59.46 | 77.04* | 80.85 | 80.32 | 83.75 | 84.22 | |
| Tropical | 75.37* | 79.14* | 82.22* | 75.89* | 75.11 | 83.79* | 83.78* | 85.41* | 85.15* | 87.61 |
| Combined | 74.55 | 79.14 | 82.22 | 75.89 | 77.04 | 83.79 | 83.78 | 85.41 | 85.15 | |

| Method | 1.4% | 3.0% | 5.0% | 11.5% | 14.5% | 16.1% | 25.9% | 30.4% | 44.5% | FM |
|---|---|---|---|---|---|---|---|---|---|---|
| | | | | | Task-5 (TREC) | | | | | |
| Standard | 24.0 | 42.2 | 56.0 | 63.0 | 64.6 | 70.2 | 96.6* | 96.8* | 97.4* | |
| Tropical | 30.8* | 45.8* | 64.6* | 71.8* | 75.8* | 73.0* | 96.4 | 96.4 | 97.2 | 97.2 |
| Combined | 30.8 | 45.8 | 64.6 | 71.8 | 75.8 | 73.0 | 96.6 | 96.8 | 97.4 | |

Table 1: Percentage of retained parameters $(100 - \hat{p})\%$ vs Test Accuracy/F1. FM refers to the full model, i.e., unpruned adapter states. Superscript '*' refers to better performing setting out of `Standard` and `Tropical`.

Given an adapter module, Algorithm 1 finds the minimizers $\hat{\mathbf{A}}$ and $\hat{\mathbf{B}}$ by performing gradient descent-based updates[3] over two loss terms expressed in terms of generators $\mathbf{G}_1$ and $\mathbf{G}_2$. $T$, $r$ denote the maximum gradient steps and the number of rows in A and columns in B. $\eta \in \mathbb{R}^+$ is step size and $\lambda_1, \lambda_2 \in \mathbb{R}^+$ indicate the importance of pruning over the shift in generators. We employ layer-wise pruning of the network without any iterative fine-tuning on downstream tasks. We find $p\%$ parameters with the smallest magnitude in $\{\mathbf{A}, \mathbf{B}\}$ and $\{\hat{\mathbf{A}}, \hat{\mathbf{B}}\}$ separately, denoted by $P_S$ and $P_T$. We denote tropical adapter pruning by `Tropical` that prunes only those parameters in $P_T$ which are also present in the set $P_S$. The final percentage of pruned parameters decreases to $\hat{p}\%$. We compare the approach with the baseline that prunes $\hat{p}\%$ of the smallest magnitude parameters from the layer. We denote this setting by `Standard`. `Combined` chooses one of `Tropical` or `Standard` whichever gives better results on the development set. We omit the comparison with `AdapterDrop`

method as even at 50% pruning, the method shows a significant drop in the performance. `Standard` inherently tests the validity of magnitude-based pruning via lottery ticket hypothesis (Wu et al., 2022) but without iterative retraining of adapter parameters. We do not iteratively fine-tune adapter parameters on the downstream task. The proposed method is agnostic to downstream tasks, models, and the learning algorithm used to train it. Thus, the framework is related to but not directly comparable to model $L_0$ sparsification (Louizos et al., 2017) and low-rank compression (Idelbayev and Carreira-Perpiñán, 2020).

## 4 Experiments

We set up a RoBERTa-base (Liu et al., 2019) with one adapter module inserted in each layer after add and layer norm sub-layer. We follow the adapter configuration from Pfeiffer et al. (2020). For pruning analysis, we consider three tasks—Emotion Recognition in Conversations (ERC), Natural Language Inference (NLI), and Text Classification (TC). For ERC, we use `MELD`, the task is to classify the emotion of an utterance given past utterances.

---

[3]Not to confuse with gradient descent used to learn model parameters. Here, it is used to solve the optimization problem in Equation (2).

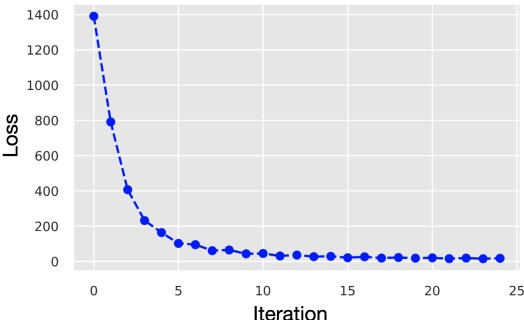

Figure 1: Value of pruning function (loss) with iterations.

| | 1.4% | 2.7% | 5.8% | 8.9% | 12.0% | 15.0% |
|------|-------|-------|-------|-------|-------|-------|
| S-CN | 71.76 | 68.48 | 65.29 | 84.52 | 85.55 | 86.96 |
| T-CN | 76.27 | 79.55 | 78.42 | 80.11 | 86.49 | 87.24 |
| S-CU | 77.49 | 74.67 | 82.65 | 86.49 | 85.18 | 87.80 |
| T-CU | 79.17 | 82.55 | 84.05 | 87.05 | 88.37 | 87.71 |
| S-CB | 69.89 | 74.39 | 58.82 | 76.17 | 85.74 | 87.90 |
| T-CB | 73.73 | 50.00 | 66.79 | 84.05 | 86.87 | 86.68 |

Table 2: Accuracy scores on RT task. Comparing node-wise (CN), layer-wise (CU), and pruning all modules together (CB). S and T denote `Standard` and `Tropical`, respectively.

Keeping the current utterance first, we append the past seven utterances in reverse order (Bhardwaj et al., 2022b). For NLI, we use SNLI dataset (Bowman et al., 2015). We append the premise and hypothesis separated by the special token . For TC task, we use three datasets: IMDB (Maas et al., 2011), Rotten Tomatoes RT (Pang and Lee, 2005), and TREC (Li and Roth, 2002). Separately, we pretrain adapters on downstream tasks with batch size 32, LR of 0.001, and 1000 steps with evaluation at every 100 steps using a development set. The evaluation metric for ERC is macro F1 and accuracy for all the other tasks. We set pruning percentage $p \in \{98\%, 96\%, \dots, 2\%\}$. Table 1 shows the test performance of networks with the percentage of adapter parameters retained, i.e., $(100 - \hat{p})\%$, this is represented in black-bold fonts. We observe that both `Standard` and `Tropical` can be effective in pruning more than 60% of the adapter parameters with a small drop in performance with respect to the full module performance (FM). Moreover, we notice `Tropical` outperforms `Standard` in eight out of nine pruned model states on MELD, six out of nine on SNLI, eight out of nine pruned adapter states on RT and IMDB, and six out of nine states on `Trec`. Across the 45 combinations of tasks and pruning fractions, except for two settings, we observe tropical geometry-based combined approach outperforms the other two, denoted in red font.

Next, we study tropical pruning in different scenarios—class-bind, class-uniform, and node-wise (See et al., 2016). In class-blind (CB), all the parameters of adapters are considered for pruning $p\%$ of the smallest magnitude weights and biases. In class-uniform (CU), we prune $p\%$ of the smallest magnitude parameters of each adapter layer separately. We also refer to it as layer-wise pruning. In node-wise pruning, we prune $p\%$ of the node-wise parameters (considering both weights and biases).

As shown in Table 2, in the `Standard` settings S-CN/ S-CU/ S-CB, we observe layer-wise S-CU pruning works best in four out of six different fractions of parameters retained. In the `Tropical` pruning settings T-CN/ T-CU/ T-CB, layer-wise pruning T-CU performs best amongst all the considered pruning fractions. Moreover, T-CU works best under each pruning fraction category.

Figure 1 shows the `Objective` function in Equation (2) quickly converges to the minimum. This observation corroborates the claim of convexity by (Alfarra et al., 2022). The plot in Figure 2 shows the change in zonotope structure before and after optimization on SNLI. The black polytope is obtained from generators $\mathbf{A}$, $\mathbf{B}$ and the red polytope shows the polytope obtained after optimization, i.e., zonotope obtained from $\hat{\mathbf{A}}$. $\hat{\mathbf{B}}$. We observe the optimization preserves the geometry of zonotopes while enforcing the rows of the down-projection matrices to be as much sparse as possible, i.e., many points in the zonotope come close to zero, keeping necessary boundary points to preserve the geometry. These zonotopes are dual to adapter hypersurfaces, thus preserving one structure enforces the other's orientation to remain preserved. Hence, one can prune adapters yet maintain their characteristic properties.

## 5   Conclusion

We proposed a novel approach for adapter pruning by studying their tropical characteristics. We formulated it as an optimization problem that aims to identify row-sparse projection matrices while minimizing the distance between tropical hypersurfaces before and after pruning. We demonstrated the advantages of tropical characterization on five NLP datasets reformulated as classification.

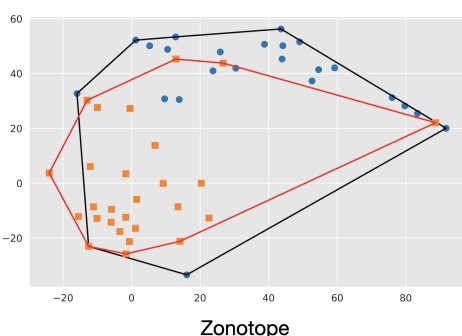

Figure 2: Zonotope defined by adapters before (red) and after the pruning (blue) via Algorithm 1.

## 6 Limitations

As our focus is on adapter-based architectures, the proposed approach can not be directly adapted to other parameter-efficient approaches such as soft prompt tuning (Lester et al., 2021; Bhardwaj et al., 2022a) which do not have explicit dense connections and activation. Another limitation comes from ReLU activation function. Since it fits in min-max (Tropical) algebra, we could reformulate the problem in terms of tropical polynomials. However, for other non-linear activation functions such as Tanh, one has to reformulate and likely resort to approximations as there is no straightforward way to cast them in a tropical algebraic expression.

## Acknowledgement

We thank the anonymous reviewers for their constructive feedback. This project is supported by the AcRF MoE Tier-2 grant (Project no. T2MOE2008, and Grantor reference no. MOE-T2EP20220-0017) titled: "CSK-NLP: Leveraging Commonsense Knowledge for NLP", and the SRG grant id: T1SRIS19149 titled "An Affective Multimodal Dialogue System".

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

## A    Tropical Algebra and Geometry

To motivate our approach we first provide background on tropical algebra and geometry.

**Tropical Arithmetic.**    Tropical algebra is a variant of classical algebra where basic arithmetic operations are redefined. The tropical sum $\oplus$ of two numbers represents their maximum and the tropical product $\odot$ represents a classical addition[4]. Thus,

$$x \oplus y = \max\{x, y\}$$
$$x \odot y = x + y$$

For instance, $2 \oplus 5 = 5$ and $2 \odot 5 = 7$. Axioms and order of arithmetic operations in tropical algebra follows the classical, thus addition is commutative and multiplication is distributive over addition:

$$x \oplus y = y \oplus z \qquad \text{(commutative)}$$
$$x \odot (y \oplus z) = x \odot y \oplus x \odot z \quad \text{(distributive)}$$

From these properties, it can be inferred that $-\infty$ is the additive identity as $-\infty \oplus x = x$ and 0 is multiplicative identity $0 \odot x = x$. Elements under the tropical arithmetic in the space of real numbers (with $-\infty$) are said to form a semiring $\mathbb{T}$ denoted by a triplet $(\mathbb{R} \cup \{-\infty\}, \oplus, \odot)$.

**Tropical Power and Monomial.**    For any variable $x \in \mathbb{T}$, the *tropical power* can be defined as $x^{\odot a} = a.x$, where $a \in \mathbb{N}$ (a natural number). For simplicity of notations, we will write $x^a$ in place of $x^{\odot a}$. A *tropical monomial* is expressed in the form

$$c\,\mathbf{x}^{\alpha} \coloneqq c \odot x_1^{a_1} \odot x_2^{a_2} \odot \ldots \odot x_n^{a_d}$$

where $c \in \mathbb{R} \cup \{-\infty\}$ and $a_i \in \mathbb{N}$. For convenience, we will write tropical monomial by $c\,\mathbf{x}^{\alpha}$ where $\mathbf{x} = (x_1, \ldots, x_d) \in \mathbb{T}^d$ and $\alpha = (a_1, \ldots, a_d) \in \mathbb{N}^d$.

**Tropical Polynomial.**    A $d$-variable tropical polynomial $f(\mathbf{x})$ can be represented by a finite sum of tropical monomials

$$f(\mathbf{x}) = c_1\mathbf{x}^{\alpha_1} \oplus c_2\mathbf{x}^{\alpha_2} \oplus \ldots \oplus c_n\mathbf{x}^{\alpha_n}$$

where the $a_i \neq a_j$ when $i \neq j$, coefficients $c_i \in \mathbb{R} \cup \{-\infty\}$, $\alpha_i = (a_{i1}, a_{i2}, \ldots, a_{id}) \in \mathbb{N}^d$ and exponents $a_i$ are integers. Ignoring $-\infty$ for ease, it is important to note that $p$ has a mapping $\mathbb{R}^d \to \mathbb{R}$, both $\mathbf{x}$ and $\alpha$ are $d$-dimensional vectors.

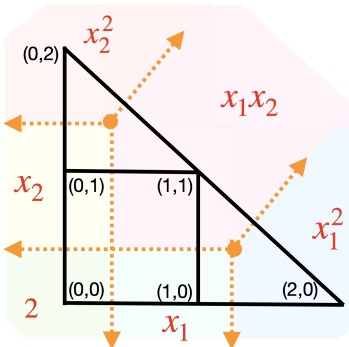

Figure 3: Tropical curve $\mathcal{F}(p)$ (orange) and dual subdivision of Newton polytope $\delta(p)$ (black) of $f(\mathbf{x}) = 1 \odot x_1^2 \oplus 1 \odot x_2^2 \oplus 2 \odot x_1 x_2 \oplus 2 \odot x_1 \oplus 2 \odot x_2 \oplus 2$.

Tropical powers, monomials and polynomials are basic building blocks of the algorithm we propose for adapter pruning.

### A.1    Tropical Hypersurfaces.

Tropical hypersurfaces are analogues to classical algebraic surfaces and key objects for us to study for adapter pruning. Given a tropical polynomial $f(\mathbf{x}) = c_1\mathbf{x}^{\alpha_1} \oplus \ldots \oplus c_n\mathbf{x}^{\alpha_n}$, its tropical hypersurface is a set of points where $p$ is attained by two or more constituting monomials, thus

$$\mathcal{F}(p) \coloneqq \{\mathbf{x} \in \mathbb{R}^d : c_i\mathbf{x}^{\alpha_i} = c_j\mathbf{x}^{\alpha_j},$$
$$\text{for some } \alpha_i \neq \alpha_j\}.$$

Here we mention a few provable facts—$\mathcal{F}$ divides the domain of $p$ into convex regions (or cells). Polynomial $p$ is non-linear at $\mathbf{x}$ if and only if $\mathbf{x}$ lies on $\mathcal{F}$. Similar to algebraic polynomials, we can identify Newton polytopes associated to tropical polynomials.

**Newton Polytopes.**    For a given polynomial $f(\mathbf{x}) = c_1\mathbf{x}^{\alpha_1} \oplus \ldots \oplus c_n\mathbf{x}^{\alpha_n}$, its newton polytope is defined by the convex hull of the exponents $\alpha_i \in \mathbb{N}^d$. The points $\alpha_i$ and polytope lies in a $d$-dimensional plane ($\mathbb{R}^d$). Thus

$$\Delta(p) \coloneqq \text{ConvHull}(\{\alpha_i \in \mathbb{R}^d : c_i \neq -\infty\}_{i=1}^n)$$

The tropical polynomial $p$ determines the dual subdivision $\delta(p)$ of newton polytope. The tropical hypersurface $\mathcal{F}(p)$ is dual graph to this $\delta(p)$, i.e., vertices of $\mathcal{F}(p)$ are regions of $\delta(p)$ and edges represent two adjacent regions in $\delta(p)$[5]. Each vertex in $\delta(p)$ corresponds to one "cell" in $\mathbb{R}^b$ where $p$

---

[4]The tropical addition can be defined as $a \oplus b = \min\{a, b\}$ or $\max\{a, b\}$, we focus on the latter as we analyze a ReLU-based adapter network.

[5]Reader can read more about dual graphs in (Deo, 2017)

is linear. Since $\delta(p)$ is in one-to-one correspondence with the tropical hypersurface, we study the adapter characteristics—underlying hypersurfaces $\mathcal{F}(p)$)—by studying the orientation of the primal graph $\delta(p)$. To determine $\delta(p)$, we use the fact that when the model is bias-free, $\delta(p) = \Delta(p)$ (Zhang et al., 2018; Alfarra et al., 2022). Figure 3 provides an illustration of $\mathcal{F}(p)$ adn $\delta(p)$ for a specific $p$.

**Zonotopes.** The zonotope formed by $\mathbf{v}_1, \ldots, \mathbf{v}_m \in \mathbb{R}^n$ is defined as $\mathcal{Z}(\mathbf{v}_1, \ldots, \mathbf{v}_m) := \{\sum_{i=1}^m \lambda_i \mathbf{v}_i, 0 \leq \lambda_i \leq 1\}$.

**Minkowski sum.** Given two sets $P_1$ and $P_2$ in $\mathbb{R}^d$, the Minkowski is defined as

$$P_1 \tilde{+} P_2 := \{\mathbf{v}_1 + \mathbf{v}_2 : \mathbf{v}_1 \in P_1, \mathbf{v}_2 \in P_2\}$$

**Property-1.** The Minkowski sum of two polytopes is the convex hull of their vertex sets. Let, $\mathcal{V}(P)$ be the vertex sets of a polytope $P$, then

$$P_1 \tilde{+} P_2 = \text{ConvHull}\left(\mathcal{V}(P_1) \tilde{+} \mathcal{V}(P_2)\right)$$

Under bias-free assumption,

**Property-2.** Let $p_1$ and $p_2$ be the tropical polynomials, then

$$\delta(p_1 \odot p_2) = \delta(p_1) \tilde{+} \delta(p_2)$$

# B Pruning Objective

## B.1 Notations Used

We denote $\mathbf{W}_d$, $\mathbf{W}_u$, $\mathbf{h}$ by $\mathbf{A}$, $\mathbf{B}$, and $\mathbf{x}$, respectively; $\mathbf{B}^+ := \max\{\mathbf{B}, \mathbf{0}\}$; $\mathbf{B}^- := \max\{-\mathbf{B}, \mathbf{0}\}$; $b_i$ denotes $i_{th}$ row of $\mathbf{B}$; $\boldsymbol{b}^{i+} := \max\{\boldsymbol{b}^i, \mathbf{0}\}$, $\boldsymbol{b}^{i-} := \max\{-\boldsymbol{b}^i, \mathbf{0}\}$; $\text{Diag}[\boldsymbol{u}]$ arranges $\boldsymbol{u}$ in a diagonal matrix; $\|\mathbf{G}\|_{1,1} := \Sigma_{k=1}^d \|\mathbf{G}(i,:)\|_1$; $\|\cdot\|_F$ denotes Frobenius Norm.

## B.2 Derivation of Pruning Objective

Let $f(\mathbf{x}) = \mathbf{B} \max\{\mathbf{A}\mathbf{x}, \mathbf{0}\}$, then

$$f(\mathbf{x}) = (\mathbf{B}^+ - \mathbf{B}^-)\left(\max\{\mathbf{A}^+\mathbf{x}, \mathbf{A}^-\mathbf{x}\} - \mathbf{A}^-\mathbf{x}\right)$$

$$= \left[\mathbf{B}^+ \max\{\mathbf{A}^+\mathbf{x}, \mathbf{A}^-\mathbf{x}\} + \mathbf{B}^-\mathbf{A}^-\mathbf{x}\right]$$

$$- \left[\mathbf{B}^- \max\{\mathbf{A}^+\mathbf{x}, \mathbf{A}^-\mathbf{x}\} + \mathbf{B}^+\mathbf{A}^-\mathbf{x}\right]$$

Thus we define $H(\mathbf{x})$ and $Q(\mathbf{x})$ as

$$H(\mathbf{x}) := \left[\mathbf{B}^+ \max\{\mathbf{A}^+\mathbf{x}, \mathbf{A}^-\mathbf{x}\} + \mathbf{B}^-\mathbf{A}^-\mathbf{x}\right]$$

$$Q(\mathbf{x}) := \left[\mathbf{B}^- \max\{\mathbf{A}^+\mathbf{x}, \mathbf{A}^-\mathbf{x}\} + \mathbf{B}^+\mathbf{A}^-\mathbf{x}\right]$$

Thus, $f(\mathbf{x}) = H(\mathbf{x}) - Q(\mathbf{x})$. Let $f^i$ denote the first output from adapter block, $b^i = \mathbf{B}[i, :]$ (i.e. $i^{\text{th}}$ row of $\mathbf{B}$). We use $\mathcal{H}$ and $\mathcal{Q}$ to denote tropical hypersurfaces of $H$ and $Q$ at node $i$.

$$\mathcal{H} = \left[\bigodot_{j=1}^p \left(\mathbf{x}^{a_j^+} \oplus \mathbf{x}^{a_j^-}\right)^{b_j^{i+}}\right] \odot \left[\bigodot_{j=1}^p \left(\mathbf{x}^{a_j^-}\right)^{b_j^{i-}}\right]$$

Computing dual subdivision

$$\delta(\mathcal{H}) = \left[\tilde{+}_{j=1}^p \left(b^{i+} \text{ConvHull}(a_j^+, a_j^-)\right)\right]$$

$$\tilde{+}\left[\tilde{+}_{j=1}^p \left(b_j^{i-} a_j^-\right)\right] \qquad \text{(P-1)}$$

$$= \left[\tilde{+}_{j=1}^p \left(b^{i+} \text{ConvHull}(a_j^+ - a_j^-, 0)\right)\right]$$

$$\tilde{+}\left[\tilde{+}_{j=1}^p \left(b_j^{i-} a_j^-\right)\right] + shift \qquad \text{(P-2)}$$

$$= \left[\tilde{+}_{j=1}^p \left(b^{i+} \text{ConvHull}(a_j, 0)\right)\right] + shift$$

Similarly, we compute dual subdivision of $q^i$

$$\delta(\mathcal{Q}) = \left[\tilde{+}_{j=1}^p \left(b^{i-} \text{ConvHull}(a_j, 0)\right)\right] + shift$$

Note that convex hull of $a_j$ and 0 is a line segment. Thus, $\delta(h^i)$ defines a Minkowski sum over line segments which is a zonotope. Following Alfarra et al. (2022), and ognoting the shifts, one can straightaway obtain zonotope generators $\mathbf{G}_1$ and $\mathbf{G}_2$ for $\delta(\mathcal{H})$ and $\delta(\mathcal{Q})$, respectibely.