# OpenReview forum: "Adapter Pruning using Tropical Characterization"
_EMNLP/2023/Conference — EMNLP 2023 Findings_

### Official Review · Reviewer_pb9m · 2023-08-02

**Typos Grammar Style And Presentation Improvements:** 1. [Line 118] In the equation of Line…
**Soundness:** 3

**Excitement:**

3: Ambivalent: It has merits (e.g., it reports state-of-the-art results, the idea is nice), but there are key weaknesses (e.g., it describes incremental work), and it can significantly benefit from another round of revision. However, I won't object to accepting it if my co-reviewers champion it.

**Paper Topic And Main Contributions:**

This paper proposes to prune the adapter with the tropical characteristics, which is formulated as an optimization problem to identify the row-sparse projection matrices. The aim is to minimize the distance of tropical hypersurfaces before and after pruning. The method is tested on five NLP classification datasets.

**Questions For The Authors:**

1. What is the essential motivation to adopt the tropical hypersurface for pruning?
2. In most cases, the adapter is small in size, is it necessary to prune the adapter?
3. Why the contribution of loss1 in Alg. 1 is set to 0.5?
4. What is the time complexity of the optimization process?

**Reasons To Accept:**

1. The paper tries to prune the adapter from the perspective of hypersurfaces.
2. The paper is evaluated on five NLP datasets.
3. This paper is clear and easy to follow.

**Reasons To Reject:**

1. The major concern about this paper is the motivation to conduct pruning on the tropical hypersurface.
2. Pruning methods have been investigated long ago; this paper does not include state-of-the-art baselines to show the comparison results.
3. Several typos and grammar issues.

**Reproducibility:**

3: Could reproduce the results with some difficulty. The settings of parameters are underspecified or subjectively determined; the training/evaluation data are not widely available.

**Reviewer Confidence:**

4: Quite sure. I tried to check the important points carefully. It's unlikely, though conceivable, that I missed something that should affect my ratings.

---

> ### Author Rebuttal · Authors · 2023-08-28
>
> **[Clarification-1]: Motivation.**
>
> - (Problem formulation): Find parameters that are least important to the network, such that removing them does not lead to significant performance degradation on the task.
>
> - (Why classical pruning is insufficient): Classical magnitude-based pruning is prone to errors as low magnitude weights can still have a huge impact on the adapter's output. Thus pruning them from the network can potentially lead to a significant drop in the model's performance. We posit that the magnitude of parameters is not sufficient to convey their importance to the network.
>
> - (Hypersurfaces to rescue): The hypersurfaces tell us about the characteristics of the function defined by the network, here the network is adapter modules. Thus, preserving the adapter's hypersurface is equivalent to preserving parameters that are important to the function, irrespective of their magnitudes. This analysis would provide extra insights into parameter importance to the network.
>
> - (How to find hypersurfaces): Because of the nonlinearity of the function, it is challenging to find exact hypersurfaces formed by the ReLU-based adapter. Thus, we adopt the zonotopes, a dual analysis that helps us identify important weights with more ease. During magnitude-based pruning, we find that preserving weights that are important for zonotope geometry retained more performance from the model than alone magnitude-based pruning of the same fraction of parameters disregarding their importance to hypersurfaces.
>
> As per the reviewer's suggestion, we plan to streamline the discussion about the paper's motivation and proposed approach in the paper.
>
> ===
>
> **[Clarification-2] Is it necessary to prune the adapter?**
>
> A task-specific adapter can take up to 10% of the overall space. When the backbone model size becomes large, consider a large language model for instance, even 10% of the extra space can be significant, thus even the parameter-efficient adapters can significantly limit the number of tasks LLMs can be specialized on. We show that the adapters can be pruned up to 80% with a little trade-off in accuracy, thus able to specialize over 5 times more number of tasks as compared to without pruning. While due to resource limitations, we could not experiment on LLMs, we provide analysis over various smaller LMs such as RoBERTa, DeBERTa, and ALBERT (please refer to Table 1 of reviewer 2vRX). Adapter pruning has seen recent interest in the research community such as structured pruning [1], adapter pruning with lottery tickets [2], and adapter-fusion pruning [3]. These works perform adapter pruning with (iterative) training, tropical pruning is directly applied to the on-the-shelf model. Thus, making [1][2][3] less interesting baselines. If the reviewer feels we should include an elaborate discussion, we will do so in the paper.
>
> ===
>
> **[Clarification3]: Why the contribution of loss1 in Alg. 1 is set to 0.5?**
>
> It's part of the expression to optimize, it denotes the factor $\frac{1}{2}$ in equation-2. The derivation can be found in the related literature [4] as well as the appendix.
>
> ===
>
> **[Clarification4]: What is the time complexity of the optimization process?**
>
> Since it is a convex optimization problem, we observe with step size $\eta$ (=0.01), the process converges to global minima in $T$ steps. Each step constitutes $r$-times gradient updates. Thus, the time complexity is $O(rT)$. Empirically, we find for $r=48$, the process converges in $T=200$ steps. We observe the optimization process is very fast, converging in about 300 seconds of time on a standard CPU machine. Thanks for pointing it out, we will include the time-complexity analysis in the paper.
>
> ==
>
> References:
>
> - [1] Hedegaard, Lukas, et al. "Structured Pruning Adapters." arXiv preprint arXiv:2211.10155 (2022).
> - [2] Wu, Jiarun, and Qingliang Chen. "Pruning adapters with lottery ticket." Algorithms 15.2 (2022): 63.
> - [3] Wu, Jiarun, et al. "Pruning Adatperfusion with Lottery Ticket Hypothesis." Findings of the Association for Computational Linguistics: NAACL 2022. 2022.
> - [4] Alfarra, Motasem, et al. "On the decision boundaries of neural networks: A tropical geometry perspective." IEEE Transactions on Pattern Analysis and Machine Intelligence 45.4 (2022): 5027-5037.

---

### Official Review · Reviewer_rtfk · 2023-08-04

**Soundness:** 4

**Excitement:**

3: Ambivalent: It has merits (e.g., it reports state-of-the-art results, the idea is nice), but there are key weaknesses (e.g., it describes incremental work), and it can significantly benefit from another round of revision. However, I won't object to accepting it if my co-reviewers champion it.

**Paper Topic And Main Contributions:**

This paper focuses on adapter pruning by studying the tropical characteristics of trainable modules. The authors propose a novel method based on tropical geometry to identify and remove irrelevant adapter parameters without altering the orientation of underlying tropical hypersurfaces. The main contribution lies in introducing a systematic optimization problem formulation for adapter pruning. Experimental results on five NLP datasets demonstrate the effectiveness of the proposed approach, which outperforms magnitude-based baselines and shows the combined approach's superiority across different tasks.

**Questions For The Authors:**

Why is it important to find adapter parameters to preserve the hypersurface geometry? Any comparison between this idea with other pruning ideas?  Is it this method to limited to adapter pruning? If not, how does this method compared to other pruning methods on other tasks?

What is the bottleneck dimension value used in this paper? Do you find any relationship between dimension and pruning ratio?

How does this method compared to the setting of finetuning the similar amount of parameters which are randomly selected?

**Reasons To Accept:**

1. The paper studies an important problem in parameter-efficient tuning methods.
2. The proposed method outperforms the magnitude-based baseline.

**Reasons To Reject:**

1. The motivation behind the design is not introduced clearly. Why is it important to find adapter parameters to preserve the hypersurface geometry? Any comparison between this idea with other pruning ideas?
2. The configuration of adapter is not clearly investigated. The bottleneck dimension is an important factor to determine the performance as well as influencing the pruning ratio. What are the bottleneck dimensions used in this setting?

**Reproducibility:**

3: Could reproduce the results with some difficulty. The settings of parameters are underspecified or subjectively determined; the training/evaluation data are not widely available.

**Reviewer Confidence:**

4: Quite sure. I tried to check the important points carefully. It's unlikely, though conceivable, that I missed something that should affect my ratings.

---

> ### Author Rebuttal · Authors · 2023-08-25
>
> **[Clarification 1]: Why find adapter parameters that preserve hypersurface geometry**
>
> - (Problem formulation): Find parameters that are least important to the network, such that removing them does not lead to significant performance degradation on the task.
>
> - (Why classical pruning is insufficient): Classical magnitude-based pruning is prone to errors as low magnitude weights can still have a huge impact on the adapter's output. Thus pruning them from the network can potentially lead to a significant drop in the model's performance. We posit that the magnitude of parameters is not sufficient to convey their importance to the network.
>
> - (Hypersurfaces to rescue): The hypersurfaces tell us about the characteristics of the function defined by the network, here the network is adapter modules. Thus, preserving the adapter's hypersurface is equivalent to preserving parameters that are important to the function, irrespective of their magnitudes. This analysis would provide extra insights into parameter importance to the network.
>
> - (How to find hypersurfaces): Because of the nonlinearity of the function, it is challenging to find exact hypersurfaces formed by the ReLU-based adapter. Thus, we adopt the zonotopes, a dual analysis that helps us identify important weights with more ease. During magnitude-based pruning, we find that preserving weights that are important for zonotope geometry retained more performance from the model than alone magnitude-based pruning of the same fraction of parameters disregarding their importance to hypersurfaces.
>
> Thus, finding parameters that are important for adapter hypersurfaces (function expressed by adapters) and preserving them during the magnitude-based pruning is highly effective at retaining model performance. As per the reviewer's suggestion, we plan to streamline the discussion about the paper's motivation and proposed approach in the provided extra space after acceptance.
>
> =======================
>
> **[Clarification 2] What are the bottleneck dimensions used in this setting?**
>
> The bottleneck dimension $r$ we used is 48. Hence an adapter block looks like the following:
> $768 \rightarrow 48 \rightarrow 768$ with ReLU activation after the down-projection. We chose $r$ as per the wide range of experiments carried out by [1]. Since the reviewer pointed out that the bottleneck dimension can be a huge contributing factor to the pruning ratio, we performed new experiments on a text classification task to investigate it. Table 1 shows full model performance with a change in $r$ and Table 2 shows the impact of $r$ on the pruning ratio.
>
> |Bottleneck dimension ($r$) |Accuracy|
> |-------------------------|--------------|
> |2|88.30|
> |8|87.36|
> |32|87.58|
> |48|87.99|
> |64|88.13|
> |128|87.89|
> |Avg|87.87 $\pm_{0.32}$|
> _Table1: Impact of $r$ on classification accuracy. Experiments were carried out on the rotten tomatoes (RT) dataset._
>
>
> |Bottleneck dimension ($r$) |% Pruning |@ 1||% Pruning |@ 5||% Pruning|@ 10|
> |-------------------------|--------------|---------------|--------|-------------------------|---------------|---------------|---------------|---------------|
> ||Standard | Tropical ||Standard |Tropical||Standard|Tropical|
> |2|68.49|68.49||78.91|78.91||83.09|**87.30**|
> |8|79.24|79.24||83.81|**92.98**||86.08|**97.65**|
> |32|75.48|**83.65**||80.94|**86.41**||90.20|**93.38**|
> |48|82.54|82.54||88.47|**94.28**||94.16|**96.96**|
> |64|75.81|**85.95**||85.05|**89.38**||88.10|**94.13**|
> |128|68.47|**71.25**||80.73|**84.17**||84.01|**87.29**|
> |Avg|73.33$\pm_{5.47}$ |   **80.16**$\pm_{6.36}$ ||    81.02$\pm_{5.17}$ |    **87.70** $\pm_{6.02}$ ||    86.10 $\pm_{4.37}$ |    **92.91** $\pm_{4.51}$|
>
> _Table 2: Percentage pruning with respect to drop in accuracy. Pruning @ x denotes the percentage of parameters pruned with 1% drop in accuracy._
>
> **Observation:** We observe that irrespective of the bottleneck dimension $r$, Tropical pruning consistently outperforms Standard pruning. However, in Table 1 with the unpruned model and Table 2 with different pruning ratios, we do not notice any consistent relation between $r$ and model performance on the task. This answers the question from the reviewer: **What is the bottleneck dimension value used in this paper? Do you find any relationship between dimension and pruning ratio?**. While such an analysis could be highly task-dependent, if the reviewer feels that this experiment should be included in the paper, we would be happy to do so.
>
> =======================
>
> **[Clarification 3] How does this method compare to the setting of finetuning the similar amount of parameters which are randomly selected?**
>
> Since our approach is applicable to off-the-shelf models and in the inference phase, we believe a more concrete baseline would be randomly pruning the same percentage of parameters. As per the reviewer's suggestion, we carried out experiments on randomly selecting the same amount of pruned parameters. In Table 3, we can observe that random pruning of parameters hurts the model performance badly, dropping it close to a random predictor i.e. ~50\%.
>
> |Bottleneck dimension ($r$) |Standard | Tropical |Random|
> |-------------------------|--------------|---------------|--------|
> |2|85.1|85.1|50.0|
> |8|84.99|88.18|65.29|
> |32|86.11|88.27|50.0|
> |48|85.45|86.39|50.5|
> |64|84.89|87.89|61.16|
> |128|82.64|85.36|50.28|
> | Avg                 | 84.66 $\pm_{1.16}$ | **87.01** $\pm_{1.12}$ | 54.54 $\pm_{6.23}$ |
>
> _Table 3: Table showing performance of a model trained on RT with pruning 80% of the adapter parameters._
>
> =======================
>
> **[Clarification-5]: Any comparison between this idea with other pruning ideas?**
>
> There are several related works such as L0 sparsification [2], low-rank compression [3], and lottery-ticket hypothesis [4][5]. We believe these are not direct baselines as they involve a critical step of tuning model parameters after compression, whereas the proposed adapter pruning, and the considered baselines directly act on an off-the-shelf trained model without further (or iterative) training of the pruned model. We discuss some of these baselines in lines 177-181. Since the reviewer pointed it out, we will include a separate section in the paper comparing the approach of adapter pruning with the relevant baselines. We extended our experiments to compare with another baseline i.e. random pruning of weights. We discuss it in Table 3 above.
>
> =======================
>
> References:
> - [1] Pfeiffer, Jonas, et al. "Adapterhub: A framework for adapting transformers." arXiv preprint arXiv:2007.07779 (2020).
> References:
> - [2] Louizos, Christos, Max Welling, and Diederik P. Kingma. "Learning sparse neural networks through
>  regularization." arXiv preprint arXiv:1712.01312 (2017).
> - [3] Idelbayev, Yerlan, and Miguel A. Carreira-Perpinán. "Low-rank compression of neural nets: Learning the rank of each layer." Proceedings of the IEEE/CVF Conference on Computer Vision and Pattern Recognition. 2020.
> - [4] Wu, Jiarun, and Qingliang Chen. "Pruning adapters with lottery ticket." Algorithms 15.2 (2022): 63.
> - [5] Wu, Jiarun, et al. "Pruning Adatperfusion with Lottery Ticket Hypothesis." Findings of the Association for Computational Linguistics: NAACL 2022. 2022.

---

### Official Review · Reviewer_28EZ · 2023-08-08

**Soundness:** 3

**Excitement:**

4: Strong: This paper deepens the understanding of some phenomenon or lowers the barriers to an existing research direction.

**Paper Topic And Main Contributions:**

The paper proposed a novel approach for adapter pruning using their tropical characteristics. The proposed method outperforms several existing baselines across multiple datasets.

**Questions For The Authors:**

1. Why standard deviation is missing in Table 1?
2. Could you explain Fig 2 better?

**Reasons To Accept:**

1) The paper tackles an important problem in NLP research - parameter-efficient learning, specifically the authors try to determine a minimal set of adapter parameters required for strong performance.
2) The proposed approach is elegant and motivated from linear algebra standpoint - preserving the tropical adapter hypersurface.


**Reasons To Reject:**

1) The mathematical motivation should be made more clear; Algo 1 box is not very well explained, some notations are not defined.
2) For the experiments, authors only compared their method with "standard" - are there more relevant baselines? If no - that's ok, but needs to be discussed.

**Reproducibility:**

3: Could reproduce the results with some difficulty. The settings of parameters are underspecified or subjectively determined; the training/evaluation data are not widely available.

**Reviewer Confidence:**

3: Pretty sure, but there's a chance I missed something. Although I have a good feel for this area in general, I did not carefully check the paper's details, e.g., the math, experimental design, or novelty.

---

> ### Author Rebuttal · Authors · 2023-08-27
>
> **[Clarification-1]: Motivation.**
>
> - (Problem formulation): Find parameters that are least important to the network, such that removing them does not lead to significant performance degradation on the task.
>
> - (Why classical pruning is insufficient): Classical magnitude-based pruning is prone to errors as low magnitude weights can still have a huge impact on the adapter's output. Thus pruning them from the network can potentially lead to a significant drop in the model's performance. We posit that the magnitude of parameters is not sufficient to convey their importance to the network.
>
> - (Hypersurfaces to rescue): The hypersurfaces tell us about the characteristics of the function defined by the network, here the network is adapter modules. Thus, preserving the adapter's hypersurface is equivalent to preserving parameters that are important to the function, irrespective of their magnitudes. This analysis would provide extra insights into parameter importance to the network.
>
> - (How to find hypersurfaces): Because of the nonlinearity of the function, it is challenging to find exact hypersurfaces formed by the ReLU-based adapter. Thus, we adopt the zonotopes, a dual analysis that helps us identify important weights with more ease. During magnitude-based pruning, we find that preserving weights that are important for zonotope geometry retained more performance from the model than alone magnitude-based pruning of the same fraction of parameters disregarding their importance to hypersurfaces.
>
> As per the reviewer's suggestion, we plan to streamline the discussion about the paper's motivation and proposed approach in the provided extra space after acceptance.
>
> ==
>
> **[Clarification-2]: Undefined terms.**
>
> While we tried to cover all the terms in the paper, if the reviewer feels the following table can enhance the readability, we are willing to include it in the paper.
>
> Notation|Definition|
> |-|-|
> |$T, r$|the maximum gradient steps and the number of rows in A and columns in B (line 150)|
> |$\eta$|step size (line 152)|
> |$\lambda_1$ and $\lambda_2$|the importance of pruning over the shift in generators (line 153)|
> |$G_1, \hat{G}_1, G_2, \hat{G}_2$|Generator matrices (line 140-141)|
> |$A, B, B^+, B^-, b^{i+}, b^{i-}, \|\|G\|\|_{1,1}$ | Specific paragraph covering common notations (line 087)|
> $\hat{A}, \hat{B}, \hat{b}^{i+}$| Defined in algorithm 1
>
> _Table1: Notations used in Algorithm 1_
>
> ==
>
> **[Clarification-3]: Why standard deviation is missing in Table 1?**
>
> Reason: There is no randomness in the process of pruning. Randomness can come from two aspects of the proposed approach:
> - Model training: Since we focus on off-the-shelf adapter-based models, we don't analyze randomness in their training.
> - Pruning: Algorithm-1 is a convex optimization problem, it has exact solutions while obtaining a new set of adapter weights. Thus, there is no randomness in the optimizing algorithm, it always converges to the global minima.
>
> However, to obtain the off-the-shelf models for pruning, we performed in-house training. We found the performance varies from 1-1.5\% across the adapter-based models and datasets. While the analysis would be less interesting to the paper as training a task-specific model is not the primary contribution, if the reviewer feels, we would be happy to include randomness in obtaining the unpruned (off-the-shelf) model.
>
> ==
>
> **[Clarification-4]: Could you explain Fig 2 better?**
>
> (Simplified explanation of how preserving zonotopes structure tends to reflect more effective pruning):
>
> Hypersurfaces tell the characteristics of the function defined by the adapters. Since directly analyzing hypersurfaces is a complex task due to highly non-linear operations, we analyze the duals of hypersurfaces using tropical algebra. Broadly, zonotopes define the dual to the hypersurfaces. Preserving zonotope structures during pruning tends to preserve the structure of the (primal) hypersurfaces, which in turn preserves the adapter properties. The plot in Figure 2 shows the change in zonotope structure before and after optimization on SNLI. The black polytope shows zonotopes before pruning and the red polytope shows the polytope obtained after pruning (Algorithm-1). These polytopes are obtained by the convex hull of the points of the corresponding color. Each point represents a row in the projection matrices A and B. We show the optimization preserves the geometry of zonotopes while decreasing the magnitude of parameters that are least important to the structure. The parameters that are not part of defining the polytope come close to zero i.e. tend to be pruned more.
>
> If the reviewer feels the discussion should be extended, we would do so in the paper.
>
> ==
>
> **[Clarification-5]: For the experiments, authors only compared their method with "standard" - are there more relevant baselines? If no - that's ok, but needs to be discussed.**
>
> Thanks for asking this. There are several related works such as L0 sparsification [1], low-rank compression [2], and lottery-ticket hypothesis [3][4]. We believe these are not direct baselines as they involve a critical step of tuning model parameters after compression, whereas the proposed adapter pruning, and the considered baselines directly act on an off-the-shelf trained model without further (or iterative) training of the pruned model. We discuss some of these baselines in lines 177-181. Since the reviewer pointed it out, we will include a separate section in the paper comparing the approach of adapter pruning with the relevant baselines. We extended our experiments to compare with another baseline i.e. random pruning of weights. We discuss it in Clarification-3 (Table 3) of reviewer rtfk.
>
> ==
>
> References:
>
> - [1] Louizos, Christos, Max Welling, and Diederik P. Kingma. "Learning sparse neural networks through $ L_0 $ regularization." arXiv preprint arXiv:1712.01312 (2017).
> - [2] Idelbayev, Yerlan, and Miguel A. Carreira-Perpinán. "Low-rank compression of neural nets: Learning the rank of each layer." Proceedings of the IEEE/CVF Conference on Computer Vision and Pattern Recognition. 2020.
> - [3] Wu, Jiarun, and Qingliang Chen. "Pruning adapters with lottery ticket." Algorithms 15.2 (2022): 63.
> - [4] Wu, Jiarun, et al. "Pruning Adatperfusion with Lottery Ticket Hypothesis." Findings of the Association for Computational Linguistics: NAACL 2022. 2022.

---

### Official Review · Reviewer_2vRX · 2023-08-12

**Soundness:** 3

**Excitement:**

4: Strong: This paper deepens the understanding of some phenomenon or lowers the barriers to an existing research direction.

**Paper Topic And Main Contributions:**

This study presents an adapter pruning approach through an examination of the tropical characteristics of trainable modules.However, I find myself somewhat puzzled by the author's reasoning behind the pruning analysis. Normally, pruning is employed to enhance model training or inference efficiency, or to improve model generalization. Nevertheless, the article does not delve into the impact of pruning on model efficiency, and the proposed method did not surpass the full model's performance in most tasks.

**Questions For The Authors:**

The code repository has expired.

**Reasons To Accept:**

- The experimental results suggest that the proposed approach could lead to enhanced performance for Pre-trained Language Model (PLM) in specific tasks.
- This study assesses the performance of PLM in relation to varying percentages of retained adapter parameters.

**Reasons To Reject:**

- The new experimental results reveal that the Computational Resources have not achieved a substantial reduction.
- The experiments inadequately discuss crucial hyperparameters, such as lambda.

**Reproducibility:**

3: Could reproduce the results with some difficulty. The settings of parameters are underspecified or subjectively determined; the training/evaluation data are not widely available.

**Reviewer Confidence:**

2: Willing to defend my evaluation, but it is fairly likely that I missed some details, didn't understand some central points, or can't be sure about the novelty of the work.

---

> ### Author Rebuttal · Authors · 2023-08-25
>
> **[Clarification-1]: This study exclusively applies its method to RoBERTa, which results in a lack of generalizability validation for the proposed approach.**
>
> We perform new experiments on DeBERTa [1] and AlBERT [2] to show that the proposed approach is generalizable across models. We observe the model to outperform the Standard baseline in 12 out of 21 settings and perform equally well in the rest of the settings.
>
> |Model                        |   1% |  3%      | 5% | 9% | 15% | 17% | 20% | FM|
> |------------------------|-------|----------|------|-----|-------|-------|-------|-----|
> |RoBERTa (Standard)| 33.56 | 35.00|41.02|39.65|49.38|50.18|55.05|60.71|
> |RoBERTa (Tropical)  | 33.56 | **37.37**|41.02|**49.59**|**52.28**|**56.34**|**58.04**|60.71|
> |DeBERTa (Standard)|34.37|40.06|40.99|45.62|48.93|49.64|62.82|62.44|
> |DeBERTa (Tropical). |34.37|40.06|**42.78**|**47.17**|**51.18**|**61.69**|62.82|62.44|
> |ALBERT   (Standard) |19.58|25.07|34.75|42.41|44.21|45.78|45.78|45.78|45.78|
> |ALBERT   (Tropical)   |19.58|25.07|**40.66**|**45.78**|**45.78**|45.78|45.78|45.78|45.78|
>
> _Table 1: Results of preserving only x% of adapter parameters trained on MELD. We approximate values to the nearest percentage of pruning as opposed to Table 1 in the paper showing the exact percentage of pruned parameters. Fraction of pruned parameters = (1-x)%._
>
> While we believe the proposed approach is strong enough to be generalizable across models and tasks, if the reviewers feel these experiments should be included in the main draft, we would do so. To prove generalizability further, we include experiments with different bottleneck dimensions of adapters in Tables 1 and 2 of reviewer-rtfk. Additionally, to make the claim even stronger, we add another baseline as random pruning in Table-3 of reviewer-rtfk.
>
> =======================
>
> **[Clarification-2]: Normally, pruning is employed to enhance model training or inference efficiency or to improve model generalization. Where does adapter pruning help?**
>
> Thanks for asking! Our method contributes to the second factor i.e. **inference efficiency**. Inference efficiency carries two aspects, one can contribute to any of the following or both:
>
> 1- Reduced computational resources used by the model for inference.
>
> 2- Increased speed of inference.
>
> _Notably, we contribute in aspect 1 i.e. **reduction of computational resources utilized by the model**_. This is particularly important for deploying models on resource-constrained devices or systems. We further elaborate on resource efficiency in Table 2.
>
> =======================
>
> **[Clarification-3]: Timing before and after pruning.**
>
> As elaborated in clarification-1, we don't contribute to the speed of inference rather we contribute to the computational resources required by the model which is proportional to the number of network parameters pruned. Since adapter pruning does not change the depth of the computational graph, it makes timing analysis less interesting since it stays the same as the standard model. Instead of timing analysis, we show the percentage reduction of the computational resources required by adapters as compared to the magnitude-based pruning:
>
> | Task | % Reduction  (Computational Resources) |
> |------|--------------------|
> | MELD | 5.9                |
> | SNLI | 13.4               |
> | RT   | 6.1                |
> | IMDB | 8.9                |
> | TREC | 14.1               |
>
> _Table2: Percentage reduction in computational resources required by the model to achieve the same pruning performance as magnitude-based pruning._
>
> If the reviewer feels that the above analysis should be a part of the main draft, we would be happy to do so.
>
> =======================
>
> **[Clarification-4]: Discussion on hyperparameter lambda.**
>
> The $\lambda$ terms denote the importance of regularizers to control the sparsity of the network in the number of line segments in zonotopes. Essentially, too big $\lambda$ values can lead to shift or distortion in hypersurfaces leading to noisy pruning. Too small $\lambda$ values will make the problem equivalent to finding another set of adapter weights but without any insights into which parameters to prune. Thus, $\lambda$ balances set the trade-off between the importance of hypersurface geometry to be preserved and sparsity in the network. Thanks for the suggestion, we will elaborate on it in the paper.
>
> =======================
>
> **[Clarification-5]: Code repository.**
>
> Thanks for pointing it out! We have made it live again.
>
> =======================
>
> References:
> - [1] He, Pengcheng, et al. "Deberta: Decoding-enhanced bert with disentangled attention." arXiv preprint arXiv:2006.03654 (2020).
> - [2] Lan, Zhenzhong, et al. "Albert: A lite bert for self-supervised learning of language representations." arXiv preprint arXiv:1909.11942 (2019).

---

### Meta-Review · Area_Chair_N3vf · 2023-09-18

**Recommendation:** 2

**Metareview:**

This paper uses concepts from Tropical geometry to identify parameters for pruning from adapters. The paper is overall well-written considering the mathematical concepts that most of the NLP community is less likely to be familiar with. Reviewers overall appreciated the elegancy of the approach, and the results. One major question that immediately comes to mind is why prune the adapters which are already supposed to be relatively small, and not prune the full model? Based on the discussion (with reviewer pb9m), it seems like the answer is that the proposed approach relies on certain geometry assumptions that are specific to adapters. This should be better explained and discussed in the paper in order to properly define the scope of the introduced solution. Overall, while this limited scope slightly reduces the excitement, it could still be of interest to other researchers in the community. The soundness should be improved by the comments from the reviewers, specifically, explaining the assumptions/ constraints and why they are suitable for adapters, and comparing to other pruning baselines.

---

### Decision · Program_Chairs · 2023-10-07

**Decision:**

Accept-Findings

**Comment:**

This paper uses concepts from Tropical geometry to identify parameters for pruning from adapters. The paper is overall well-written considering the mathematical concepts that most of the NLP community is less likely to be familiar with. Reviewers overall appreciated the elegancy of the approach, and the results. One major question that immediately comes to mind is why prune the adapters which are already supposed to be relatively small, and not prune the full model? Based on the discussion (with reviewer pb9m), it seems like the answer is that the proposed approach relies on certain geometry assumptions that are specific to adapters. This should be better explained and discussed in the paper in order to properly define the scope of the introduced solution. Overall, while this limited scope slightly reduces the excitement, it could still be of interest to other researchers in the community. The soundness should be improved by the comments from the reviewers, specifically, explaining the assumptions/ constraints and why they are suitable for adapters, and comparing to other pruning baselines.